# The Study of the Kinetics of Metalaxyl Accumulation and Dissipation in Durian (*Durio zibethinus* L.) Leaf Using High-Performance Liquid Chromatography (HPLC) Technique

**DOI:** 10.3390/plants10040708

**Published:** 2021-04-06

**Authors:** Supawadee Phetkhajone, Aussanee Pichakum, Wisuwat Songnuan

**Affiliations:** 1M.Sc. Programme in Plant Science, Faculty of Graduate Studies, Mahidol University, Nakhon Pathom 73170, Thailand; supawadee.phet22@gmail.com; 2Department of Plant Science, Faculty of Science, Mahidol University, Bangkok 10400, Thailand; aussanee.pic@mahidol.ac.th; 3Department of Pharmaceutical Botany, Faculty of Pharmacy, Mahidol University, Bangkok 10400, Thailand

**Keywords:** dissipation, durian, foliar spray, HPLC, metalaxyl, soil drench

## Abstract

Metalaxyl is an effective approach to control *Phytophthora palmivora* infection in durian plantation. However, inappropriate metalaxyl usage may increase production cost, pathogen with fungicide resistance, and environmental toxicity. This study established and validated a simple and reproducible procedure to measure metalaxyl concentration in the durian leaf using HPLC. Linearity of the detection ranged from 1–100 µg/mL. The limits of detection (LOD) and quantification (LOQ) were 0.27 and 0.91 µg/mL, respectively. The extraction method gave recovery rates ranging from 88% to 103%. Durian seedlings were treated with 4 g/L metalaxyl either by foliar spray or soil drench. The highest metalaxyl accumulation in durian leaf was found between 6–24 h after treatment and persisted above its effective concentration at least 60 days after foliar application. The dissipation pattern fit to a first-order kinetics equation showed a half-life of 16.50 days. Soil drenching led to eight times higher metalaxyl concentrations in plants than foliar spraying and caused plant death within 15 days after application. These results suggest that foliar spraying of 4 g/L metalaxyl or soil drenching at a lower concentration every two months is sufficient in controlling *P. palmivora* infection in durian seedlings.

## 1. Introduction

Durian (*Durio zibethinus* L.) is one of the most sought-after Southeast Asian fruits with its unique taste and aroma. Despite the highly economic values, especially for Thailand, durian cultivation of the popular commercial cultivars, such as Monthong, faces a momentous problem of root and stem rot disease [1]. This disease is caused by *Phytophthora palmivora*, an oomycete pathogen that causes devastating diseases not only in durian, but also in various other plants such as cocoa, orchid, oil palm, and eucalyptus [2]. The pathogen can infect all plant parts, and the infection can start from young seedlings to a full-grown tree. Virulent isolates of the pathogen can cause tree death in less than a week, even before the tree starts to produce its first fruit. Therefore, *P. palmivora* control strategies must be applied from a young age.

Existing strategies to control *P. palmivora* are breeding cultivars with resistance, good cultural practice, biological control, and fungicide usage, especially Bordeaux mixture, phosphonates, and metalaxyl [3]. Most of these fungicides are systemic fungicides with acropetal or basipetal movement, which could effectively target soilborne pathogens by translocating inside the plant after application.

Fungicide penetration and dissipation into plants are complex processes associated with a myriad of factors from the fungicide itself, plant-related factors, and other environmental factors [4]. Physicochemical properties of fungicides can be exemplified by lipophilicity (octanol/water partition coefficient) [5]. Plant-related factors include the leaf cuticle, which is the main barrier of fungicide penetration, leaf surface, moisture status, and metabolic activity. Lastly, environmental conditions, such as temperature, humidity, rainfall, and wind speed, can influence fungicide penetration and dissipation.

Metalaxyl (methyl-*N*-(2,6-dimethylphenyl)-*N*-(2-methoxyacetyl)-DL-alaninate) is a systemic acylanilide fungicide used to control plant diseases caused by oomycete pathogens such as *Phytophthora*, *Pythium*, and downy mildew. It was first launched in 1977 by the Ciba-Geigy Corporation and has been used commercially since 1979 [6]. The biochemical mode of action of metalaxyl is the inhibition of fungal RNA synthesis [7,8]. Metalaxyl was reported to translocate in the xylem with a low lipophilic property (logK_ow_ value of 1.6) [9,10], which implied that it can slowly penetrate through the leaf cuticle and rapidly transport inside the plant. However, the addition of surfactant can increase the penetration ability of metalaxyl [11]. After penetration into the plant, metalaxyl accumulates and gradually dissipates systemically by translocating upward with the transpiration stream as well as being degraded [12]. There are several approaches to apply metalaxyl such as foliar spraying, soil drenching, and trunk painting. Intensive use of metalaxyl to control *P. palmivora* in the durian industry has led to the development of metalaxyl-resistant isolates in conjunction with other negative environmental impacts.

Current practice for metalaxyl application after visible symptoms is to use trunk painting and/or soil drenching until the tree is recovered. In the case of a severe outbreak, soil drenching is applied repeatedly until the outbreak is subdued. Moreover, several orchard owners routinely use metalaxyl as a preventative measure by foliar spraying metalaxyl at the recommended concentration every three to seven days. These cultural practices are rarely based on scientific evidence about the effectiveness and negative biological and environmental impacts.

To reduce the risk of metalaxyl overuse, more information about metalaxyl residue and efficacy inside durian plants is required. Analytical methods to detect metalaxyl residue in other plants had been previously reported, such as high-performance liquid chromatography (HPLC) [13,14,15,16], ultra-performance liquid chromatography (UPLC) [17], liquid chromatography tandem mass spectrometry (LC–MS/MS) [18,19,20], high-performance liquid chromatography tandem mass spectrometry (HPLC-MS/MS) [21], and gas chromatography (GC) [22]. The dissipation patterns of metalaxyl were studied in grape, pepper, potato, tomato, cucumber, and Swiss chard plants [13,14,16,18,21] with the half-life ranging from 0.4–6.0 days. However, there is no report of metalaxyl dissipation and degradation in durian.

The purpose of this study was to evaluate the dissipation pattern and half-life of metalaxyl in durian leaf under a greenhouse condition for the management of *P. palmivora* disease in the seedling stage. This was accomplished by establishing a simple, accurate, and precise extraction method coupled with an HPLC technique to detect metalaxyl in the durian leaf. These data are crucial for determining metalaxyl concentration and frequency of application that is more effective in controlling *P. palmivora* in the durian nursery with less negative biological and environmental impacts.

## 2. Results and Discussion

### 2.1. Optimization of Metalaxyl Extraction from Durian Leaf Sample

To precisely determine the kinetics of metalaxyl in durian seedlings, an optimized protocol for metalaxyl extraction from the durian leaf had to be developed. durian leaf was the chosen plant part not only because of its availability, ease of extraction, but also due to metalaxyl accumulation, which is the greatest in this organ [23]. Four organic solutions, known as dichloromethane, ethyl acetate, acetonitrile, and methanol, were evaluated for their efficacy of metalaxyl extraction from durian leaf samples. Extraction efficacy was assessed by spiking 0.5 g of untreated durian leaf samples with 100 µL of 100 µg/mL metalaxyl standard. Of the four solvents, ethyl acetate was the most efficient solvent at extracting metalaxyl from durian leaf tissues with a recovery rate of 100.3%, which is followed by methanol (93.6%), acetonitrile (79.9%), and dichloromethane (66.4%) (Figure 1A). Additionally, ethyl acetate has a higher evaporation rate than the other solvents, except for dichloromethane, resulting in the more rapid extraction protocol. Therefore, ethyl acetate was chosen for further analyses.

To optimize the extraction volume, spiked samples were extracted using 5, 10, or 20 mL of ethyl acetate for 500 mg of durian leaf tissue. The results showed that increasing the solvent volume did not increase the recovery rate of metalaxyl (Figure 1B), and, thus, 5 mL of ethyl acetate was chosen as the optimal extraction volume. The decreasing recovery rate with increasing extraction volume was likely due to the more diluted samples being injected into the HPLC system, reaching the limit of quantitation (LOQ) and resulting in under-detection. In an attempt to further concentrate the metalaxyl extract and remove impurities, C18 solid-phase extraction (SPE) was tested based on the extraction protocol from Velkoska-Markovska et al. [15], except that acetonitrile (ACN) was used for column conditioning and elution. However, the amount of metalaxyl recovered was below the detection limit. In addition, the quality and quantity of metalaxyl recovered using our developed method was sufficient and the use of the cartridge was not absolutely required. To simplify the method, we decided to exclude the use of the cartridge.

In the foliar spray experiment, metalaxyl residue on the leaf surface could result in a high background reading and interfere with the experiment. To determine how much residue remained on the surface of the leaf, metalaxyl concentration from an uncleaned leaf was compared to that of a leaf cleaned with running tap water one day after foliar spray application of 70 mL of 4 g/L metalaxyl per seedling. The metalaxyl concentration in an uncleaned leaf was approximately 125.8 ± 34.4 µg/g tissue, whereas, in a cleaned leaf, it was only 22.7 ± 3.7 µg/g tissue (Figure 1C). This result suggests that a significant level of metalaxyl is deposited on the leaf surface. Therefore, the surface cleaning step was added to the extraction protocol.

Four concentrations of acetonitrile in water: 0%, 25%, 50%, and 100% were tested to determine the most suitable solvent for re-dissolution. Untreated leaf samples were spiked with 100 µL of 100 µg/mL of metalaxyl before extraction and dissolution with the described protocol. All of the tested acetonitrile solutions had comparable efficacy to re-dissolve metalaxyl from the durian leaf crude extract (Figure 1D). The 100% acetonitrile solvent was selected to re-dissolve dried extracts based on metalaxyl solubility, minimal preparation time, and good peak separation in the chromatogram.

In summary, the optimized protocol for metalaxyl extraction from the durian leaf is as follows: 0.5 g cleaned leaf tissue extracted with 5 mL ethyl acetate and re-dissolved with 100% acetonitrile without further clean-up (Figure 1E). Previously reported metalaxyl extraction and preparation protocols for grape, pepper, cucumber, potato, tomato, and Swiss chard [13,14,16,17,18,21] used larger amounts of plant tissues (10–25 g) and larger volumes of extraction solvents (40–100 mL). Moreover, the clean-up methods required specific chemical compounds and equipment. In comparison, this extraction protocol required a simple method with minimal sample weight, small volume of extraction solvent, and no complex clean-up method.

### 2.2. HPLC Method Validation

HPLC was used to detect metalaxyl in the present study. The chromatogram of metalaxyl extracted under optimal conditions indicated high specificity with a retention time of 3.828–3.998 min with no interfering peak, as shown in Figure 2. The method was validated for linearity, LOD, LOQ, accuracy, and precision using an external standard of metalaxyl and matrix-matched experiment for metalaxyl quantification. The calibration curve based on six concentrations of the metalaxyl standard (0.5, 1, 5, 10, 50, and 100 μg/mL) showed satisfactory linearity and regression coefficients (R^2^) at 0.999 with LOD and LOQ of 0.15 and 0.51 µg/mL, respectively. For the matrix-matched experiment with six concentrations of identical metalaxyl spiked into untreated leaf samples, the signal could not be obtained from the lowest concentration of 0.5 µg/mL. The linearity of spiked durian leaf samples ranged from 1–100 µg/mL with a regression coefficient (R^2^) of 0.999. LOD and LOQ were 0.27 and 0.91 µg/mL, respectively (Table 1). Previously reported LOD and LOQ of metalaxyl detection using the HPLC technique ranged from 0.0015 to 0.015 ppm while LOQ ranged from 0.001 to 1 ppm [13,14,15,16,17,19]. The linearity, LOD, and LOQ detected in the solution and in the matrix of crude durian leaf extract were highly similar, suggesting that the matrix did not significantly interfere with the extractability of metalaxyl.

To determine the recovery rate, accuracy, and precision, 100 µL of metalaxyl standard solutions at 1, 10, and 100 µg/mL was spiked into untreated durian leaf samples, extracted, and analyzed three times intra-day for three consecutive days (inter-day). The average percent recovery of metalaxyl in durian leaf ranged from 84.9–105.9 with %RSD between 0.4–14.6 and 88.1–103.4 with %RSD of 3.4–9.8 when the analyses were performed intra-day and inter-day, respectively (Table 2). Percent recovery was calculated from the following equation: (100 × (detected concentration/spiked concentration)). According to the percent recovery equation, there are several reasons why the recovery rate could be higher than 100%. For example, degradation/precipitation of the metalaxyl standard or impurities in the sample could cause the detected signals of the standard to be lower or signals of the samples to be slightly higher than expected, which led to a positive error. In such cases, the recovery rate exceeding 100% could be observed. Nonetheless, this percent recovery (103%) is well within the acceptable range following the previous described guideline (70–120%). The developed method was considered satisfactory in terms of repeatability and reproducibility based on a previously established guideline [24].

### 2.3. Metalaxyl Concentration 24 h after Foliar Spray Application

The effect of surfactant on penetration efficacy of metalaxyl into the durian leaf was assessed by measuring metalaxyl concentrations 24 h after spraying. The results showed that metalaxyl concentrations in durian leaf were 14.60 ± 1.91 and 6.38 ± 0.73 µg/g tissue, when applied with and without 0.1% Tween^®^20, respectively (Figure 3A). These data suggest that metalaxyl spray supplemented with Tween^®^20 resulted in approximately two times higher metalaxyl concentration *in planta*.

As expected, the uptake of metalaxyl by durian seedling increased as the applied concentration increased (Figure 3B). The detected concentrations of metalaxyl were 1.62, 2.76, 7.23, and 15.56 µg/g leaf tissue, after foliar spray application at 0.5, 1, 2, and 4 g/L metalaxyl, respectively. Furthermore, metalaxyl accumulation was separately investigated for each section of the seedling: basal, middle, and apical parts from 0 to 24 h after foliar spray application (Figure 4). The results showed that, for all three sections, initial metalaxyl concentrations were approximately 2.37 ± 0.21 µg/g tissue and increased significantly (up to 19.82 ± 1.68 µg/g tissue) after 6 h. Based on the average concentration of detected metalaxyl (about 20 µg/g tissue) and the total leaf mass of about 80 g per seedling, the total *in planta* amount of metalaxyl was approximately 1600 µg. Assuming that 50% of the sprayed metalaxyl was lost as run-off, the total amount of applied metalaxyl was approximately 35,000 µg/seedling. Thus, the maximum penetration of metalaxyl into durian seedling detected in this study was 4.6%. No further increase was observed from 6 h to 24 h after application. Metalaxyl accumulation appeared to be uniform in all plant parts at all time points. No further increase was observed from 6 h to 24 h after application. Metalaxyl accumulation appeared to be uniform in all plant parts at all time points.

The penetration efficacy of fungicide into plants is associated with leaf surface characters, fungicide physicochemical properties, environmental conditions, and the addition of surfactants [25]. In this study, it was found that the nonionic surfactant Tween^®^20 helped fungicide to penetrate into durian leaves that have thick and waxy cuticles, possibly due to its cellular plasmalemma reaching property and increased cuticular transport process of fungicide. Therefore, it is recommended that foliar spray of metalaxyl should be applied with the surfactant to increase metalaxyl efficacy. In addition, surfactants could also increase the long-term stability of agrochemicals [25]. However, this aspect was not investigated in the current study. The 4.6% uptake of metalaxyl in durian was comparable to the previous report using radioactive labeling techniques, which detected metalaxyl uptake inside *Persea indica* and tomato at around 2–5% [23]. Better metalaxyl penetration through leaf sheath was found in maize seedlings at about 10% of applied metalaxyl [26]. More than 90% of applied metalaxyl remained on the leaf blade surface [23,26,27].

Metalaxyl movements have been shown to involve translocation in the xylem from the application site to the apical part of plants through the transpiration system [9,12]. Zaki et al. [23] reported that metalaxyl was translocated to the apical parts of *Persea indica* at 14 days after foliar spraying. Similar results that metalaxyl accumulated on the upper part of plants and mostly accumulated at the edge of the leaf blade were found in maize and tomato seedlings [9,26]. However, this study did not find clearly higher accumulation in the apical part when compared to middle and basal parts. This could be due to several reasons. First, the durian tissues were observed 24 h after application in this study, which may have been too early to detect accumulation. Secondly, several studies had reported metabolic action occurring inside the plants, including ring methyl hydroxylation, aryl hydroxylation, ester cleavage, O-dealkylation, and N-dealkylation [12,28,29,30]. It is possible that these processes are more active in the apical part of durian seedlings, thereby, masking the accumulation.

In this study, the *in planta* metalaxyl concentration was shown to have a linear correlation with the applied concentration in the range of 0.5–4 g/L and time after application from 0–24 h. Similarly, a linear increase of metalaxyl concentration was found in 100-fold dilution series from 0.041–4.9 µg/mL and between 3–9 h after application in tobacco [31]. Although higher concentration of applied metalaxyl led to a higher concentration *in planta*, there is no evidence that the application of the fungicide beyond 4 g/L is beneficial.

### 2.4. Metalaxyl Dissipation in Durian

As previously mentioned, metalaxyl concentration reached the maximum concentration within 6 h after foliar spray application. To investigate dissipation in the durian leaf, a longer time-course experiment (up to 60 days) was performed. A 48.1% decrease of metalaxyl inside durian leaves was observed from day 1 (16.46 ± 1.61 µg/g tissue) to day 7 (8.54 ± 0.83 µg/g tissue). Metalaxyl persisted on durian leaf for 60 days after application at a concentration of 3.44 ± 0.39 µg/g tissue, or 20.9% of the highest residue (Figure 5A, Table 3). Previously, Chan and Kwee [32] reported that metalaxyl at an *in planta* concentration of 1 µg/mL was enough to control mycelial growth, sporangium formation, and chlamydospore production of *P. palmivora* in durian. Therefore, the current terminal concentration of metalaxyl after the foliar spray should still be sufficient in controlling root and stem rot disease caused by *P. palmivora* in durian.

In comparison, Yang et al. [13] reported that 98% of metalaxyl was degraded within three days after foliar spray application to potato plants under field conditions. Similarly, metalaxyl dissipated rapidly after one day of application and decreased 79.2–92.8% after seven days of application in pepper plants [17]. The slower dissipation rate found in this study might have been because of the lower activity in durian, the lower sunlight level in the greenhouse when compared to the field condition, or the irrigation method. In general, the degradation of fungicides could be influenced by plant activity and environmental conditions such as light, temperature, and relative humidity [14]. The high persistence of metalaxyl had been previously reported in citrus plants such as tangelo and lemon trees up to 138–160 days after the trunk painting application [33]. Further experiments should be performed with full-grown durian trees to ascertain that the metalaxyl dissipation rate is low in durian, and, thus, frequent fungicide application is not needed.

To broaden the knowledge of metalaxyl dynamics in durian, another application method known as soil drenching was studied. One day after application, the detected metalaxyl concentration in durian leaf was minimal (2.25 µg/g tissue). At the same applied concentration of 2 L of 4 g/L metalaxyl as with the foliar spray method, soil drenching led to the *in planta* concentration of 127.40 ± 10.44 µg/g, or approximately eight times higher than that of foliar spray application at seven days after application (Figure 5B, Table 3). Fifteen days after the soil application, metalaxyl likely accumulated to a concentration too high for the durian seedling to tolerate, resulting in plant death. It has been previously reported that metalaxyl soil drench had higher effectiveness to control *Phytophthora* crown rot in peach for 60 days while metalaxyl as a foliar spray was ineffective to control this disease [34]. Although it was unexpected that soil drenching of metalaxyl at the recommended concentration was toxic and caused the death of seedlings, it could, perhaps, be explained. Firstly, the seedlings used in this study were much smaller, and, thus, was likely to be more sensitive to the fungicide than the average durian trees in the plantations. Secondly, the soil drench application method used in this study was to soak the pots of seedlings in metalaxyl to the full holding capacity in order to obtain uniform results, which likely led to more metalaxyl being trapped in the rhizosphere than if the seedlings were planted in the soil where metalaxyl could leach out in all directions. The dissipation of metalaxyl in the durian leaf was fitted to the first-order kinetics equation: C_t_ = 9.108e^−0.042t^ with R^2^ of 0.90 for foliar spray method. The half-life of metalaxyl was calculated from the degradation rate constant (k) of the regression equation. The metalaxyl half-life value when applied as a foliar spray was 16.50 days. The dissipation equation could not be determined for the soil drench application because durian seedlings died after only two time points. In comparison, the half-life of metalaxyl in other plants, such as potato, tomato, cucumber, grape, pepper, and Swiss chard, were considerably lower than in durian, ranging between 0.8–6.0 days [13,14,16,17,18,21]. Although it is not surprising that metalaxyl half-life values in different plants were dissimilar, the reason for a much longer half-life value in durian was unclear. Perhaps one possibility is that durian is a woody tree with high persistence of fungicide in plants. Further studies of metalaxyl dissipation in other woody trees should be undertaken to better understand the genuine range of dissipation and half-life of metalaxyl. Overall, our results indicated that metalaxyl at the concentration of 4 g/L applied as foliar spray to control *P. palmivora* could persist at effective concentration within the plants for at least two months. Furthermore, soil drench application with the same metalaxyl concentration could lead to a much higher *in planta* concentration and could accumulate to a toxic level for seedlings. Therefore, lower concentrations are recommended for the soil drench application.

## 3. Materials and Methods

### 3.1. Chemicals and Reagents

For HPLC analysis, the metalaxyl standard with ≥98.0% purity was purchased from Sigma-Aldrich^®^, St. Louis, MO, USA (CAS. No. 57837-19-1) and prepared as a 1000 μg/mL standard stock solution in acetonitrile (ACN). Commercial-grade metalaxyl wettable powder (WP) product with 25% active ingredients, obtained from Sahaphol Kemekaset Ltd., (Bangkok, Thailand), was used for durian seedling treatments. HPLC-grade ACN and analytical-grade dichloromethane, ethyl acetate, formic acid, methanol, and Polysorbate (Tween^®^20) were purchased from Merck (Darmstadt, Germany). The deionized water used for HPLC analysis was purified using the Milli-Q^®^ Advantage A10 Water Purification System (Merck Millipore, Darmstadt, Germany).

### 3.2. Plant Materials

Durian seedlings were purchased from a plant nursery in Chumphon province, Thailand. Monthong cultivar scion was grafted onto a rootstock of local cultivar at approximately one month after germination and further cultivated in plastic pots (6.5-inch diameter, 7.5-inch soil depth) for one year after grafting. Seedlings were approximately 90–100 cm in height, with 4–6 branches and around 5–10 leaves per branch at the time of the experiments. Seedlings were maintained in a greenhouse with good irrigation and were never exposed to metalaxyl prior to the experiments.

### 3.3. Metalaxyl Treatments

Commercial-grade metalaxyl (25% a.i. WP) was applied by foliar spraying or soil drenching at the concentration of 4 g/L (1 g/L active ingredient). Metalaxyl powder was mixed in RO water with 0.1% Tween^®^20 for foliar spray application, and without Tween^®^20 for soil drench application. For foliar spraying, the suspension was manually sprayed with a spray bottle onto the adaxial leaf surface until all leaves were completely covered and excess had run off (approximately 70 mL/plant). Soil drenching was achieved by submerging the durian seedling pots into 2 L of metalaxyl suspension. After 5 min, pots were removed and drained until individual drops were observed before returning to the greenhouse. To ensure that samples in each time-course experiment were uniformly processed, seedlings were treated at separate time points and leaf samples were collected and processed at the same time. The experiments were performed in a greenhouse with a completely randomized block design (CRBD). Each treatment was performed in triplicates and the experiment was repeated three times independently. Six to nine leaves were sampled from three seedlings.

### 3.4. Leaf Sample Extraction

Leaves of metalaxyl-treated seedlings were collected and freshly extracted for HPLC analysis. The leaf surface was cleaned with running tap water for approximately 10 s per leaf before being cut into small pieces (approximately 5 × 5 mm in size), pooled, and weighed to be 500 mg/sample. Each sample was ground to a fine powder using liquid nitrogen and extracted with 5 mL of ethyl acetate overnight. One milliliter of supernatant was taken to a fresh 2 mL microcentrifuge tube and left to dry at room temperature. The dry pellet was redissolved with 200 µL of ACN and the suspension was filtered through a 0.45 µm nylon syringe filter into a 1.5 mL vial for HPLC analysis.

### 3.5. HPLC Analysis

Metalaxyl analysis was performed by WATERS 2695 Separations Module HPLC system with Empower^TM^ software equipped with a sample manager, a quaternary solvent manager connected to UV detector (Waters Co., MA, USA) and SunFire^TM^ C18, and a 5 µm column (4.6 mm × 150 mm) (Waters Co., MA, USA). Ten microliters of the sample were injected using the auto-injector with a 100 µL injection loop for the total analytical time of 12 min. The mobile phase was ACN: 0.1% formic acid in water and a flow rate of 1 mL/min with gradient elution. The following mobile phase gradient was used: 0–2 min, 55% ACN, 2–5 min, ACN was increased from 55% to 65%, 5–7 min held at 65% ACN, 7–7.5 min, ACN was decreased from 65% to 55%, and 7.5–12 min, held at 55% ACN. The detection wavelength was set at 220 nm.

### 3.6. Method Validation

Metalaxyl standards (0.5, 1, 2.5, 5, 10, 50, 100, and 500 μg/mL) were analyzed to determine linearity, limits of detection (LOD), and quantification (LOQ), accuracy, and precision of metalaxyl analysis by HPLC. Calibration curves were established by plotting the peak areas versus metalaxyl concentrations. The limit of detection (LOD), defined as the minimum amount of analyte that can be detected with a signal to noise ratio (S/N) of 3, and limit of quantification (LOQ), defined as the minimum amount of analyte that can be quantified with S/N of 10 (Kabir et al., 2018), were calculated by Empower^TM^ software. Precision and accuracy of the methods were evaluated by a recovery assay using spiked samples at three concentrations (1, 10, and 100 μg/mL) of metalaxyl in durian.

### 3.7. Calculation of Metalaxyl Concentration and Dissipation

Percent dissipation of metalaxyl was calculated using the following equation: [100 − (A_t_ × (100/A_0_))], where A_0_ is the initial residue and A_t_ is the residue at t days after metalaxyl application. The estimated concentrations of metalaxyl were calculated based on the established calibration curve by Empower^TM^ software. Calculation of the dissipation pattern followed the first-order kinetic equations: C_t_ = C_0_e^−kt^. Half-life was calculated using the following equation: t_1/2_ = (ln 2)/k, where C_t_ (μg/mL) is the metalaxyl concentration at time t (d), C_0_ (μg/mL) is the initial concentration, k is the rate constant, and t_1/2_ is the metalaxyl half-life. All data were statistically analyzed using PASW statistics version 18.0. (SPSS Inc., Chicago, IL, USA, 2009).

## 4. Conclusions

A simple, accurate, and reproducible procedure was developed for metalaxyl detection in the durian leaf with satisfactory quality and quantity. Leaf samples were extracted with ethyl acetate and analyzed using the HPLC system with the C18 column. Good separation chromatograms were achieved using gradient elution of acetonitrile–0.1% formic acid. The present protocol is satisfactory in terms of linearity, accuracy, and precision. This procedure was applied to show that metalaxyl concentration in durian leaves peaked after 6 h with no difference between basal, middle, and apical parts of the plant. The concentration of metalaxyl started to decrease after 1 day but continued to persist on durian leaves for 60 days after application with a half-life of 16.50 days. On the other hand, soil drenching with metalaxyl resulted in toxic concentrations, which the seedlings could not tolerate. In summary, our results suggest that metalaxyl foliar application at a concentration of 4 g/L is sufficient in controlling *P. palmivora* for two months and soil drench application at the same concentration is not advised. Lower concentration of metalaxyl is suggested for soil drench application for the management of root and stem rot disease in durian seedling. These results can help in establishing a guideline for the safe and suitable application of metalaxyl in durian orchards. Although this study focused on developing a metalaxyl detection procedure and metalaxyl kinetics in durian seedling, the obtained information should also be beneficial for future studies, especially studies relating to metalaxyl detection and kinetics in mature durian trees or an edible part of durian fruits and other products from durian.

## Figures and Tables

**Figure 1 plants-10-00708-f001:**
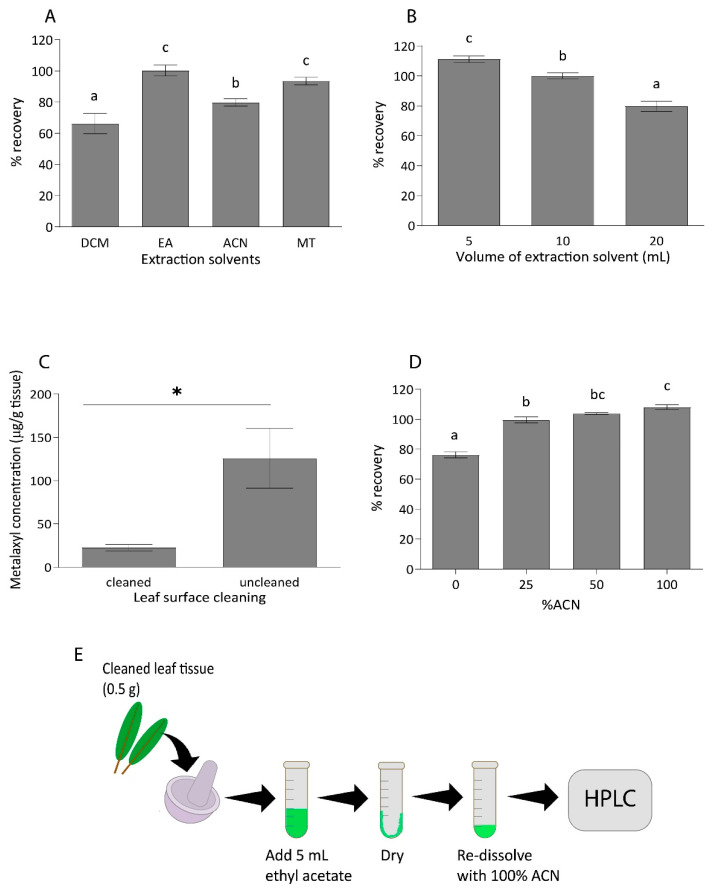
Metalaxyl extraction protocol development. Average percent recovery of metalaxyl in durian leaf extracted with four organic solvents: dichloromethane (DCM), ethyl acetate (EA), acetonitrile (ACN), and methanol (MT) (**A**), and with three extraction volumes: 5, 10, and 20 mL (**B**). Durian leaf samples were spiked with 100 µL of metalaxyl standard before extraction. Metalaxyl concentrations on cleaned and uncleaned leaf samples (**C**). Metalaxyl concentration after re-dissolution with 0%, 25%, 50%, and 100% ACN (**D**). (**E**) A summary diagram showing the developed protocol for metalaxyl extraction from durian leaf. Error bars with different letters show a significant difference at a significant level of 0.05. Data in (**A**,**B**,**D**) were tested with one-way ANOVA with Tukey’s HSD post-hoc test (*n* = 3). The experiment in (**C**) was tested with a paired *t*-test (*n* = 3). The asterisk (*) indicates statistically significant difference at *p* < 0.05.

**Figure 2 plants-10-00708-f002:**
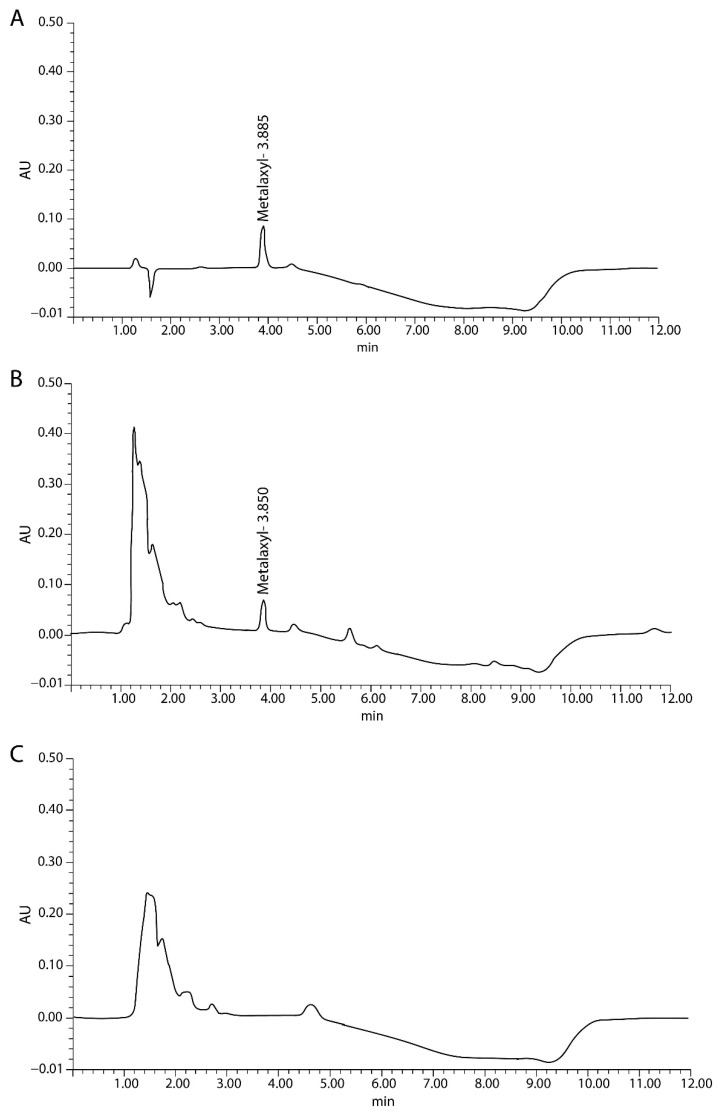
Chromatogram of 10 µg/mL metalaxyl standard, (**A**) 10 µg/mL spiked durian leaf sample, (**B**) and untreated durian leaf sample (**C**).

**Figure 3 plants-10-00708-f003:**
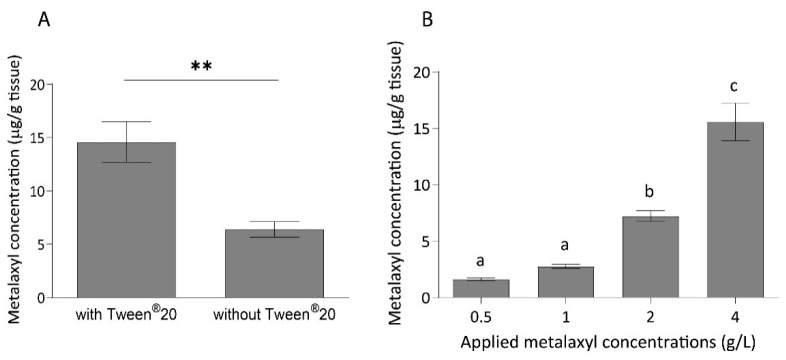
Effect of surfactant and applied concentration on metalaxyl *in planta* concentrations. (**A**) Metalaxyl concentrations in durian leaf 24 h after spraying with 4 g/L metalaxyl with and without 0.1% Tween^®^20. (**B**) Metalaxyl concentrations in durian leaf 24 h after spraying with 0.5, 1, 2, and 4 g/L metalaxyl. Data are mean ± standard error (SE) among three independent experiments. The experiment in (**A**) was tested with a paired *t*-test (*n* = 5), while (**B**) was tested with one-way ANOVA with Tukey’s HSD post-hoc test (*n* = 27). The bars with the same letter designated no significant difference at the significant level of 0.05. The double asterisk (**) indicates statistically significant difference at *p* < 0.05.

**Figure 4 plants-10-00708-f004:**
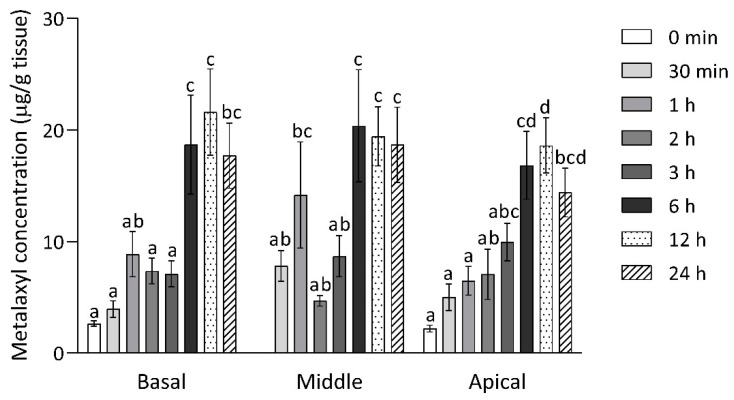
Metalaxyl concentrations in basal, middle, and apical parts of durian seedlings. The error bars represent the standard errors of nine replications from a pooled sample of 6–9 leaves from three seedlings. The bars in the same time point with the same letter designate no significant difference at *p* < 0.05 using ANOVA with Duncan’s multiple range test.

**Figure 5 plants-10-00708-f005:**
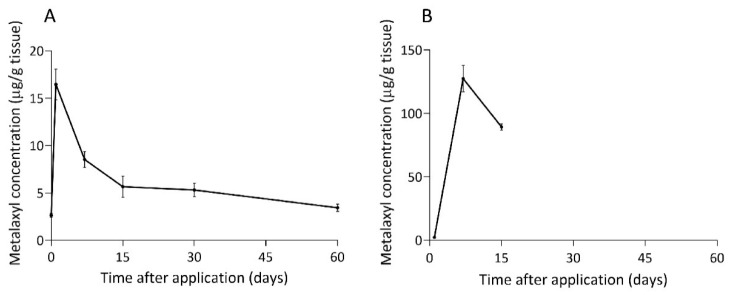
Metalaxyl concentrations in durian leaf after spraying at the concentration of 4 g/L foliar spraying (**A**) and soil drenching (**B**). Because durian seedlings treated with drenching died within 15 days, data collection was discontinued. Each bar represented the standard error of means (SEM) from nine replications at a significant level of 0.05 using ANOVA followed by Duncan’s multiple range test. Two independent experiments were performed.

**Table 1 plants-10-00708-t001:** Linear equation, correlation coefficient (R^2^), LOD, and LOQ of metalaxyl (0.5–100 µg/mL) in acetonitrile solvent and spiked leaf sample.

Analyte	Matrix	Linear Equation	Linear Range (µg/mL)	R^2^	LOD	Linear Range (µg/mL)
Metalaxyl	Acetonitrile	y = 32,787x–9550.5	0.5–100	0.999	0.15	0.51
	Durian leaf	y = 36,621x–653.63	1–100	0.999	0.27	0.91

**Table 2 plants-10-00708-t002:** Average recovery of metalaxyl in durian leaf for repeatability and reproducibility determination.

Spiked Level (µg/mL)	Intra-Day		Inter-Day ^1^	
	Average (*n* = 6)	%RSD	Average (*n* = 18)	%RSD
1	85.36	2.79	88.07	9.77
	84.87	14.56		
	93.99	4.10		
10	98.98	0.41	103.40	4.06
	105.31	2.92		
	105.91	2.96		
100	98.24	0.60	102.13	3.42
	102.40	0.88		
	105.74	1.76		

%RSD represents the percentage of relative standard deviation. ^1^ Experiments were run on three consecutive days.

**Table 3 plants-10-00708-t003:** Average recovered concentrations of metalaxyl in durian leaf for repeatability and reproducibility determination.

Application Method	Days after Application	Average Residue ± SE	%Dissipation
Foliar spray	0	2.65 ± 0.21	N/A
	1	16.46 ± 1.61	0
	7	8.54 ± 0.83	48.10
	15	5.67 ± 1.14	65.55
	30	5.53 ± 0.73	67.69
	60	3.44 ± 0.39	79.08
Soil drench	0	BDL	BDL
	1	2.25 ± 0.47	N/A
	7	127.40 ± 10.44	N/A
	15	89.31 ± 2.55	N/A
	30	N/A	N/A
	60	N/A	N/A

BDL: below the detectable limit. N/A: not applicable because seedlings died prior to data collection date.

## Data Availability

Not applicable.

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
