# Peer review of "The Study of the Kinetics of Metalaxyl Accumulation and Dissipation in Durian (Durio zibethinus L.) Leaf Using High-Performance Liquid Chromatography (HPLC) Technique"

_plants, 2021, doi:10.3390/plants10040708_

Round 1

Reviewer 1 Report

The paper is devoted to the study of metalaxyl accumulation and dissipation patterns in durian leaf. To establish the kinetics of accumulation and dissipation of metalaxyl in durian leaf, the authors created a simple method for measuring metalaxyl concentration in durian leaf using high-performance liquid chromatography (HPLC) with ultraviolet (UV) detection. Some of the obtained results are interesting. However, there are several unclear and questionable points. I recommend the paper for publication if the authors clearly address the following issues:

  1. Page 1, lines 2-4: The article title should be improved because the ‘kinetics’ word can be applied only concerning process but not to chemical substance. That title as follows can be used: “The study of the kinetics of metalaxyl accumulation and dissipation in durian leaf using high-performance liquid chromatography”.
  2. Page 3, lines 105-107 and figure 1B: The authors should explain why increasing the solvent volume did not increase the recovery rate of metalaxyl.
  3. Page 3, lines 108-111: Why did the authors not use other cartridges for solid-phase extraction, for example, ones with aminopropyl silica or Hypercarb type adsorbents?
  4. Page 5, lines 168-169, Table 2: The authors should explain the recovery rate values exceeded 100%. The formula for the calculation of the recovery rate value should be provided.
  5. Page 7, lines 212-213: Expressly, the lipophilicity of metalaxyl can not increase due to the presence of surfactant, but its presence can cause the Rebinder’s effect well-known in colloid science.
  6. Page 12, lines 375-377: The authors should choose what mode they used during HPLC: isocratic or gradient. If they used gradient elution, mobile phase composition could not be constant, as indicated in lines 375 and 405.
  7. Page 12, line 378: The model or trademark of the used sample injector for HPLC should be provided. The volume of the sample loop of the used sample injector for HPLC also should be provided.
  8. Page 12, line 385: Have the authors recalculated metalaxyl equilibrium concentrations in acetonitrile solution with considering the volume of the sample loop of the used sample injector for HPLC during calibration curve establishing?
  9. Page 13, lines 405-406: The composition of the mobile phase can not be foxed during gradient elution (see item 6).

Author Response

Reviewer 1

The paper is devoted to the study of metalaxyl accumulation and dissipation patterns in durian leaf. To establish the kinetics of accumulation and dissipation of metalaxyl in durian leaf, the authors created a simple method for measuring metalaxyl concentration in durian leaf using high-performance liquid chromatography (HPLC) with ultraviolet (UV) detection. Some of the obtained results are interesting. However, there are several unclear and questionable points. I recommend the paper for publication if the authors clearly address the following issues:

  1. Page 1, lines 2-4: The article title should be improved because the ‘kinetics’ word can be applied only concerning process but not to chemical substance. That title as follows can be used: “The study of the kinetics of metalaxyl accumulation and dissipation in durian leaf using high-performance liquid chromatography”.

         Thank you for the comment. The title was changed as suggested.

  1. Page 3, lines 105-107 and figure 1B: The authors should explain why increasing the solvent volume did not increase the recovery rate of metalaxyl.
    In our protocol, we used 500 mg of durian leaf tissue and 5 ml of solvent for the extraction. This tissue:solvent ratio is within comparable range used in previous studies (Liu et al., 2014; Yang et al., 2015; Velkovska-Markovska et al., 2017). First, since the recovery rate resulting from extraction with 5ml solvent was already ~100%, increasing the recovery rate with increasing solvent volume was unlikely. On the contrary, the increased solvent volume appeared to decrease the recovery rate. We believe this was due to the fact that the 10 ml and 20 ml extraction volume resulted in a more diluted sample injected into the HPLC system (because only 1 ml of each extraction was concentrated for the HPLC injection). The diluted samples from 10 ml and 20 ml extraction volume produced the metalaxyl peaks that were close to the LOQ, and were likely to be under-detected. 

    Since the 5ml extraction volume resulted in satisfactory recovery rate with the least drying time, we believe that the larger extraction volume was not necessary.

    To clarify this issue, we added "The decreasing recovery rate with increasing extraction volume was likely due to the more diluted samples being injected into the HPLC system, reaching the limit of quantitation (LOQ) and resulting in under-detection." to line 107-110

  2. Page 3, lines 108-111: Why did the authors not use other cartridges for solid-phase extraction, for example, ones with aminopropyl silica or Hypercarb type adsorbents?
    Based on the literature, C18 was the most frequently used adsorbent (Liu et al., 2014; Wu et al., 2017; Velkovska-Markovska et al., 2017) for similar assays. To clarify this point, we added “In addition, we felt that “the quality and quantity of metalaxyl recovered using our developed method was sufficient and that the use of the cartridge was not absolutely required. To simplify the method, we decided to exclude the use of the cartridge.” To line 114-116

  3. Page 5, lines 168-169, Table 2: The authors should explain the recovery rate values exceeded 100%. The formula for the calculation of the recovery rate value should be provided.
    To clarify this point, we changed line 174-182 to “Percent recovery was calculated from the following equation: (100 × (detected concentration/spiked concentration)). According to the percent recovery equation, there are several reasons why the recovery rate could be higher than 100%. For example, degradation/precipitation of the metalaxyl standard or impurities in the sample could cause the detected signals of the standard to be lower or signals of the samples to be slightly higher than expected, which led to the positive error. In such cases, the recovery rate exceeding 100% could be observed. Nonetheless, this percent recovery (103%) is well within the acceptable range following the previous described guideline (70-120%)”

  4. Page 7, lines 212-213: Expressly, the lipophilicity of metalaxyl can not increase due to the presence of surfactant, but its presence can cause the Rebinder’s effect well-known in colloid science.
    We agree with this point and changed line 221-226 to “The penetration efficacy of fungicide into plants is associated with leaf surface characters, fungicide physicochemical properties, environmental conditions, and the addition of surfactants [24]. In this study, it was found that the nonionic surfactant Tween®20 helped fungicide to penetrate into durian leaves that have thick and waxy cuticles, possibly due to its cellular plasmalemma reaching property and increased cuticular transport process of fungicide.”

  5. Page 12, lines 375-377: The authors should choose what mode they used during HPLC: isocratic or gradient. If they used gradient elution, mobile phase composition could not be constant, as indicated in lines 375 and 405.
    The gradient elution was used as described in line 396-400 as follows "The mobile phase was ACN: 0.1% formic acid in water (65:35 v/v), flow rate 1 ml/min with gradient elution. The following mobile phase gradient was used: 0–2 min, 55% ACN; 2-5 min, ACN was increased from 55% to 65%; 5–7 min held at 65% ACN; 7–7.5 min, ACN was decreased from 65% to 55%; 7.5–12 min, held at 55% ACN." To clarify this point, we changed line 428-429 to “Good separation chromatograms were achieved using gradient elution of acetonitrile-0.1% formic acid.”

  6. Page 12, line 378: The model or trademark of the used sample injector for HPLC should be provided. The volume of the sample loop of the used sample injector for HPLC also should be provided.
    An autoinjector was attached to the HPLC machine: WATERS 2695 Separations Module. To clarify this point, we modified line 395-396 to “The samples were auto-injected with the sample loop volume of 100 µl.”

  7. Page 12, line 385: Have the authors recalculated metalaxyl equilibrium concentrations in acetonitrile solution with considering the volume of the sample loop of the used sample injector for HPLC during calibration curve establishing?
    We did not calculate the equilibrium concentrations. Since our injector was the partial loop injection, then keeping the injection volume constant is a veritable and advisable option. Also, because the samples were dissolved in ACN before injection, and the injection volume was constant, the equilibrium to ACN in the sample loop should be internally controlled and only led to minor differences during calibration curve establishment. Because the calibration curve equation was satisfactory and within acceptable range, we believe that the established calibration curve was sufficiently reliable for the determination of metalaxyl concentration in the durian leaf. Any minor effects from the equilibrium should not change the major findings and conclusion of the experiment.

  8. Page 13, lines 405-406: The composition of the mobile phase can not be foxed during gradient elution (see item 6).
    The lines 428-429 were modified to be “Good separation chromatograms were achieved using gradient elution of acetonitrile-0.1% formic acid”.

Reviewer 2 Report

Clarify sampling, appears to be a single plant for each treatment with multiple samples collected over time. Would like the software used for statistical analysis to be identified. Add a chromatogram to show the chromatography. Would like to know what the resolution is for any adjacent peaks. Table 4 seems unnecessary as there is only the one result. Summarize in a sentence in the text and delete. The soil drench result seems surprising. A discussion of the treatment seems warranted for appropriateness. Using the van't Hoff equation the water potential associated with the drench is -0.35 Atmospheres which seems extremely high for a tropical plant. Did the plant appear water stressed?

Additional comments:

What is the main question addressed by the research?

The kinetics of metalaxyl applied as foliar application or soil drench to Durio.

Is it relevant and interesting?

Kinetics are interesting.  Only 1/2 the study produced results. No results for the soil drench. Study design appears not address kinetics for drench which is a common method of applying metalaxyl.

How original is the topic?

Metalaxyl is commonly applied to Durio sp. the dose they used seems inconsistent with common pratice for the soil drench as it proved toxic.  This should be expanded on.

What does it add to the subject area compared with other published material?

Foliar kinetics is interesting. Method of assessing systemic levels could be explained in the context of other methods used for this purpose.  

Is the paper well written?

Grammatically okay. The audience of the paper is uncertain as not enough for a pesticide chemistas missing chromatography. Not enough for a horticulturist as missing relevant practice. No microbiology evaluated either.

Is the text clear and easy to read?

Need to clarify numbers of plants per treatment group.  n=1? 

Are the conclusions consistent with the evidence and arguments presented?

For foliar data have a kinetic response to treatment.  Methodology is unusual for a systemic fungicide. 

Do they address the main question posed?

Only provide results for the foliar application.  The soil drench could be removed from the manuscript as no results provided. This is probably a major revision if done.

Author Response

Reviewer 2

Clarify sampling, appears to be a single plant for each treatment with multiple samples collected over time.

To clarify this point, we have added text “Each treatment was performed in triplicates and the experiment was repeated three times independently. Six to nine leaves were sampled from three seedlings” in line 377-379

Would like the software used for statistical analysis to be identified.

To clarify this point, we added the information about statistical analysis software in line 422-423 as  “All data were statistically analyzed using PASW statistics version 18.0. (SPSS Inc, Chicago, 2009).

Add a chromatogram to show the chromatography. Would like to know what the resolution is for any adjacent peaks.

The sample chromatograms were provided in Figure 2. The metalaxyl peak was well-resolved with little shouldering. We could not provide the identity of each adjacent peak because only the metalaxyl external standard was available.

Table 4 seems unnecessary as there is only the one result. Summarize in a sentence in the text and delete.

We agree with this point and have deleted Table 4. To summarize this information, we changed the text in line 310-316 to be “The dissipation of metalaxyl in the durian leaf was fitted to the first-order kinetics equation: Ct = 9.108e-0.042t with R2 of 0.90 for foliar spray method. The half-life of metalaxyl was calculated from the degradation rate constant (k) of the regression equation. The metalaxyl half-life value when applied as a foliar spray was 16.50 days. The dissipation equation could not be determined for the soil drench application because durian seedlings died after only two time points.“

The soil drench result seems surprising. A discussion of the treatment seems warranted for appropriateness. Using the van't Hoff equation the water potential associated with the drench is -0.35 Atmospheres which seems extremely high for a tropical plant. Did the plant appear water stressed?

The plants did not seem to experience water stress. It appears that there are some misunderstandings here. First, the applied concentration was 4g/l of metalaxyl powder, which contained only 25% active ingredient, as described in the methods line 366-367 "Commercial-grade metalaxyl (25% a.i. WP) was applied by foliar spraying or soil drenching at the concentration of 4 g/l (1 g/l active ingredient)". At this applied concentration of ~3.6 mM, the water potential of the solution would be -0.0089 and not -0.35 Atm based on the van't Hoff equation. Also, the drench was only applied for five minutes before the seedlings were removed and irrigated normally afterwards, as described in line 371-374 "Soil drenching was achieved by submerging the durian seedling pots into 2 liters of metalaxyl suspension. After 5 minutes, pots were removed and drained until individual drops were observed before returning to the greenhouse". This procedure should not exert too much osmotic stress on the seedlings.

Additional comments:

What is the main question addressed by the research?

The kinetics of metalaxyl applied as foliar application or soil drench to Durio.

Is it relevant and interesting?

Kinetics are interesting.  Only 1/2 the study produced results. No results for the soil drench. Study design appears not address kinetics for drench which is a common method of applying metalaxyl.

Even Though the soil drench experiment was terminated early, we felt that the data obtained in this experiment were still informative in at least two aspects. First, this experiment provided evidence that soil drenching could lead to much higher in planta metalaxyl concentrations than foliar spraying in young durian seedlings. Second, young seedlings were likely to be more sensitive to the toxicity of metalaxyl than full-grown trees.

How original is the topic?

Metalaxyl is commonly applied to Durio sp. the dose they used seems inconsistent with common pratice for the soil drench as it proved toxic.  This should be expanded on.

Metalaxyl soil drench is one of the common cultural practices in durian plantations. The manufacturer's recommendation provided only the concentration of metalaxyl when applied as soil drench, but did not provide the volume of usage. Once metalaxyl is applied to soil, a large proportion is likely to leach to deeper soil layers. However, plants were directly soaked into the metalaxyl solution in a pot in this study because we believe that soaking the soil with the metalaxyl solution to its full holding capacity would be the method that gives the most uniform results. As a consequence, it is likely that the plants got more metalaxyl from soaking than regular drenching from the beginning. Moreover, the metalaxyl trapped inside the pot is less likely to be drained out by irrigation compared to that in the soil. The fact that the seedlings used in this study were much smaller, and thus was likely to be more sensitive to the fungicide, than average durian trees in the plantations likely exacerbated the situation.

To expand the explanation, we added "Although it was unexpected that soil drenching of metalaxyl at the recommended concentration was toxic and caused the death of seedlings, it could perhaps be explained. Firstly, the seedlings used in this study were much smaller, and thus was likely to be more sensitive to the fungicide than average durian trees in the plantations. Secondly, the soil drench application method used in this study was to soak the pots of seedlings in metalaxyl to the full holding capacity in order to obtain uniform results, which likely led to more metalaxyl being trapped in the rhizosphere than if the seedlings were planted in the soil where metalaxyl could leach out in all directions. To line 303-310

What does it add to the subject area compared with other published material?

Foliar kinetics is interesting. Method of assessing systemic levels could be explained in the context of other methods used for this purpose.  

So far, other previous studies aiming to assess kinetics of metalaxyl inside several species of plants that we came across all used comparable methods of analysis (Liu et al., 2012; Liu et al., 2014; Ramezani and Shahriari, 2015; Yang et al., 2015; Malhat, 2017; Kabir et al., 2018). Thus, we believe that our methods should suffice, and we are unsure of what other methods of systemic metalaxyl kinetics analysis should be explained further.

Is the paper well written?

Grammatically okay. The audience of the paper is uncertain as not enough for a pesticide chemistas missing chromatography. Not enough for a horticulturist as missing relevant practice. No microbiology evaluated either.

Thank you for the comment. Being less specific about one particular area could also mean that the information in this manuscript is accessible by a broader audience. We initiated this study because we believed there was not enough literature that could serve as a reference for the development of a better Phytophthora palmivora management program. Reliable metalaxyl kinetics data provided in this manuscript and a few others from our lab in the near future could lead to the reduction of fungicide use and a more sustainable agricultural practice with a minimal negative impact on the environment.

Is the text clear and easy to read?

Need to clarify numbers of plants per treatment group.  n=1? 

To clarify this point, we have added text “Each treatment was performed in triplicates and the experiment was repeated three times independently. Six to nine leaves were sampled from three seedlings” in line 377-379

Are the conclusions consistent with the evidence and arguments presented?

For foliar data have a kinetic response to treatment.  Methodology is unusual for a systemic fungicide. 

Again, thank you for the comment. The HPLC analysis of systemic fungicides is a standard practice in our experience.

Do they address the main question posed?

Only provide results for the foliar application.  The soil drench could be removed from the manuscript as no results provided. This is probably a major revision if done.

Although the soil drench data provided were limited to the dissipation in the first 15 days after metalaxyl application, we believe it is still a highly valuable information. In particular, few other studies reported comparisons of different metalaxyl application methods along with the actual concentrations inside the plants. In our opinion, leaving out the soil drench data would surely reduce the impact of the manuscript.

Round 2

Reviewer 1 Report

The article has been significantly improved after the authors have clearly addressed notes 1-5, 8-9 from my previous review and have edited appropriate parts of their article. But unfortunately, there are some issues that should be corrected before the article will have been published. First, it is difficult to find out the injection volume used in work because in lines 405-406, the authors indicate that the sample loop volume was 100 μl, while in line 410, it is told that injection volume was 10 μl. This contradiction should be eliminated. Secondly, solvent composition 65:35 v/v indicated in line 407 should be removed because later the authors describe gradient elution details (lines 408-410) that contradict to information provided in line 407. After performing these revisions, the article can be published in Plants.

Author Response

Response to reviewer’s comments:

Reviewer 1

The article has been significantly improved after the authors have clearly addressed notes 1-5, 8-9 from my previous review and have edited appropriate parts of their article. But unfortunately, there are some issues that should be corrected before the article will have been published.

First, it is difficult to find out the injection volume used in work because in lines 405-406, the authors indicate that the sample loop volume was 100 μl, while in line 410, it is told that injection volume was 10 μl. This contradiction should be eliminated.

To clarify this point, we changed line 405-407 to be “Ten microliters of the sample were injected using the auto-injector with 100 µl injection loop for the total analytical time of 12 min.”

Secondly, solvent composition 65:35 v/v indicated in line 407 should be removed because later the authors describe gradient elution details (lines 408-410) that contradict to information provided in line 407. After performing these revisions, the article can be published in Plants.

The issue was changed as suggested. (line 407)